# Damage Generated and Propagated by the AAR Reactive Aggregate from Kingston, Ontario, Canada

**DOI:** 10.3390/ma17010166

**Published:** 2023-12-28

**Authors:** Cassandra Trottier, Leandro F. M. Sanchez

**Affiliations:** Department of Civil Engineering, Faculty of Engineering, University of Ottawa, Ottawa, ON K1N 6N5, Canada; leandro.sanchez@uottawa.ca

**Keywords:** alkali–aggregate reaction, alkali–carbonate reaction, internal swelling reaction, multi-level assessment, damage rating index, semi-quantitative microscopy

## Abstract

It remains unclear in the literature what the cause of the so-called alkali–carbonate reaction (ACR) damage to concrete is. However, expansion and cracks as distress features are often attributed to the alkali–silica reaction (ASR). Therefore, this work aims to assess the damage to concrete generated and propagated by the so-called ACR-susceptible reactive aggregate through mechanical testing (i.e., the direct shear test), microscopy (the damage rating index—DRI), and other techniques. Distinct induced expansion levels (i.e., 0%, 0.05%, 0.12%, and 0.20%) were selected to compare the distress caused by ACR to concrete affected by ASR. The results show that the behavior of ACR, namely, as captured through the DRI, is inconsistent with that of ASR, thus attesting to ACR being a distinct distress mechanism. The damage captured through mechanical testing does not distinguish ACR from ASR; however, microscopy reveals that cracks in the cement paste are the main damage mechanism. The proportions of cracks in the cement paste are 40–50% of the total number of cracks, whereas open cracks in the aggregates normally characterizing ASR represent only up to 20% of the total cracks.

## 1. Introduction

The alkali–aggregate reaction (AAR) is a harmful distress mechanism in concrete, resulting in a reduction in its performance and leading to a shortened service life. AAR can be divided into two main types of mechanisms: the alkali–silica reaction (ASR) and the so-called alkali–carbonate reaction (ACR); the former is more widely spread and understood. However, ACR in concrete presents many challenges since the mechanism of expansion and deterioration is not yet fully understood. In ASR-affected concrete, cracks begin within the aggregate and extend into the cement paste as the expansion increases. On the other hand, ACR-affected concrete is subjected to dedolomitization, which turns the dolomite into brucite and calcite (Equation (1)). However, these processes are said to not cause expansion or expansion-related cracks [1], while it is believed that the mechanism causing cracks in such concrete is the presence of ASR [1,2,3]. Dedolomitization (Figure 1b) is considered a harmless process by some researchers [1,4] although the so-called ACR can cause significant expansion and damage to concrete [5,6]. ACR can therefore be characterized by dark rims on the aggregate’s boundary (also visible in ASR-affected concrete) and a white carbonate halo around the aggregate in the cement paste (Figure 1a).
(1)$$\begin{array}{c} CaMg{\left(CO_{3}\right)}_{2}+2OH_{aq}^{-}\to Mg{\left(OH\right)}_{2}+CaCO_{3}+CO_{3}^{2-}{}_{aq}\\ \mathrm{dolomite}\;\;\;\;\;\mathrm{brucite}\;\;\mathrm{calcite} \end{array}$$

Most studies currently focus on revealing ACR’s mechanism through a variety of testing methods. However, ACR continues to be present in concrete infrastructure, and its distinction from other mechanisms is necessary through mechanical testing and microscopy. In a previous study by Sanchez et al. [8], an evaluation of the cause and extent of damage through multi-level assessment was conducted. This entails a combination of mechanical testing and microscopic observations. Tests able to capture cracking from a diagnostic perspective are limited, such as the stiffness damage test (SDT) and modulus of elasticity [9,10], along with the understanding of the influence of cracking on aggregate interlock [11,12]. As such, using the damage rating index (DRI) for various types of AAR-reactive aggregates, it was found that the so-called ACR-reactive aggregate from Kingston, Ontario, produced a completely different cracking behavior and pattern when compared to ASR-reactive aggregate. It remains unclear whether ASR is the only mechanism responsible for such behavior. Meanwhile, it was found that some mechanical property losses in the so-called ACR-affected concrete are in accordance with losses observed in ASR-affected concrete. Cracking in concrete affects its mechanical properties and permeability, and thus, its structural performance, serviceability, and further durability. Such cracks and representative ACR features are observed at scales that correspond to engineering properties. Nevertheless, the DRI is a semi-quantitative microscopy tool that uses a stereomicroscope at 15–16x magnification while the petrographer/operator counts petrographic/distress features in the field of view of 1 cm by 1 cm on the surface of a polished/reflective concrete section (Figure 2). The DRI was developed to establish areas of relative damage in a structure (i.e., a dam affected by ASR) based on the counted features, thus producing a semi-quantitative evaluation of the damage [13] influenced by [14]. Each feature count is multiplied by a weighting factor that represents the importance of the petrographic/distress feature towards the overall observable/measurable damage. The sum of all counts is set to 100 cm^2^ for comparative purposes and to account for differences in core/specimen sizes. Due to the various types of feature counts, bar charts can be generated to visualize the importance of certain features. The original method was used namely as a petrographic tool [13,14,15,16,17]; however, further developments attest to the DRI’s engineering capabilities by relating mechanical properties to the observed counts when using the weighting factors that provide the least variability [8,18,19]. Cracks observed at the meso-scale and quantified through counts can further be used to model damage developments [20,21].

This work therefore aims to compare the distress mechanism of concrete made with Kingston coarse aggregate to ASR-affected concrete through crack generation and propagation to better understand the so-called ACR-affected concrete and its influence on mechanical property losses. Moreover, a description of the damage generated by ACR will be provided to allow stereomicroscopy operators to identify features caused by ACR and distinguish them from ASR, allowing for optimization of the diagnostic process of stereomicroscopy. Therefore, in this study, concrete made with the Kingston reactive coarse aggregate will be evaluated mechanically, microscopically, and through other techniques at distinct expansion levels (i.e., 0%, 0.05%, 0.12%, and 0.20%) to understand the damage generated as a function of the expansion level when compared to ASR-affected concrete. The study encompasses a reference reactive aggregate that has been under evaluation for several decades [1,6,22,23,24] and does not generalize among all ACR-susceptible aggregates.

## 2. Materials and Methods

### 2.1. Concrete Specimen Manufacturing and Monitoring

A total of sixty cylindrical (i.e., 10 cm in diameter by 20 cm in length) concrete specimens were manufactured in a laboratory using ACR-reactive coarse aggregate from Kingston, Ontario, combined with a local non-reactive fine aggregate (Table 1). The mixture proportions followed those of the concrete prism test (CPT) as per ASTM C1293, using a Type 1 GU cement (i.e., 0.86% Na_2_O_eq_) at 420 kg/m^3^ and a water-to-cement ratio of 0.45 (Table 2). The total alkali content of the concrete mixture was raised to 5.25 kg/m^3^ and 1.25% Na_2_O_eq_ by cement mass, using reagent-grade NaOH to accelerate the reaction. After fabrication, the concrete specimens were left to moist cure for 24 h (i.e., 100% RH and 20 °C), after which the specimens were demolded. Stainless steel studs were then installed in both ends of the specimens, where holes (i.e., 8.5 mm in diameter and 19 mm in length) were drilled using a press drill equipped with a masonry drill bit. The studs were fastened using a quick-setting cement paste slurry, and the concrete specimens were left to cure for an additional 24 h under the same aforementioned conditions. The initial zero readings were then taken using a digital micrometer, and specimens were stored under conditions enabling the reaction (i.e., 100% RH and 38 °C). Length-change measurements were taken periodically to monitor the expansion over time.

### 2.2. Mechanical Testing: Direct Shear Resistance Test

The direct shear resistance of the concrete specimens was captured using the set-up proposed by [12] and more recently adapted to evaluate the effect of AAR with respect to the expansion level achieved [27,28]. Specimens were circumferentially cut at a depth of 21 mm [12] using a masonry saw equipped with a notched diamond blade. The notch width was equal to the blade’s width of 5 mm, which ensures maximum resistance. The applied loading rate was selected to be 100 N/s since rates lower than this value did not influence the shear resistance, whereas faster loading rates resulted in an increase in the shear strength. The apparatus used is shown in Figure 3, and the maximum force at which the concrete specimen failed was converted into shear resistance using Equation (2), as follows:(2)$$\tau =\frac{4P}{\pi {\left(\phi -2a\right)}^{2}},$$
where *P* is the failure load (N), ϕ is the diameter of the cylinder (mm), and *a* is the depth of the notch (mm).

### 2.3. Microscopy: The Damage Rating Index (DRI)

The DRI was performed on the affected concrete specimens, which were cut in half longitudinally in one motion using a masonry saw equipped with a notched diamond blade to reduce the amount of heat generated and with water used as a coolant. The flat surfaces were ground and polished using a mechanically rotating polishing table and magnetic laps from coarse to fine grits of 30, 60, 140, 280 (80–100 μm), 600 (20–40 μm), 1200 (10–20 μm), and 3000 (4–8 μm). Between each lap, the specimens were cleaned of loose debris, then dried with compressed air to remove loose particles from the surface and avoid excessive scratching of the reflective surface. A 3D-printed grid with squares of 1 cm by 1 cm was placed on the reflective surface, and each square was evaluated using a stereomicroscope at 16× magnification (i.e., field of view of the 1 cm^2^ square) by counting the observed distress features in the form of cracks. Weighting factors were then applied to the types of distress features encountered (Table 3), which were selected based on their influence on the concrete’s properties, whereby cracks in the cement paste have a higher weighting factor due to the severity of such distress. A lower weighting factor was attributed to features that least represented damage, while higher weighting factors represented more advanced stages of damage and significant losses in mechanical responses. The sum of all weighted counts was then normalized to 100 cm^2^, providing the DRI number (Equation (3)). More than 100 cm^2^ per specimen was evaluated to ensure that the DRI result was statistically significant, which will be graphically illustrated with the results.
(3)$$DRI=\left[\frac{\sum (0.25\left(CCA\right)+2\left(OCA+OCA_{RP}+DAP\right)+3\left(CCP+CCP_{RP}+CAD\right))}{\mathrm{Number}\;\mathrm{of}\;\mathrm{analyzed}\;1\;\mathrm{cm}\;\mathrm{by}\;1\;\mathrm{cm}\;\mathrm{squares}}\right]\times 100\;{\mathrm{cm}}^{2}$$

### 2.4. Other Techniques: Apparent Porosity

The apparent porosity can be described as the open capillary voids/pores within the concrete in which a liquid can intrude while under pressure. The apparent porosity was determined using the Archimedes immersion method, as described by [29], and calculated using Equation (3).
(4)$$P=\frac{V_{instrusion}}{V_{object}}=\frac{m_{SSD}-m_{D}}{m_{SSD}-m_{Sub}},$$

First, the concrete specimens were cut into three equal disks using a masonry saw equipped with a notched diamond blade and washed to remove any remaining debris. The initial mass of the disks was recorded, and the disks were placed in an oven at 60 °C until they were dried at a constant mass, m_D_. The disks were immersed in a water bath equipped with a vacuum pump, and the lid was sealed to ensure the pressure remained constant. The pump was turned on for 3 min or until all air had been removed from the concrete disks (bubbling had ceased) and left to submerge into the vacuum-sealed water bath for 24 h. The mass in SSD condition, m_SSD_, was achieved by removing the concrete disks from the water bath and drying the surface with a dry cloth, after which the disks were placed in the submerged balance to record the submerged mass, m_Sub_.

## 3. Results

### 3.1. AAR Expansion over Time

The average expansion as a function of time is shown in Figure 4, where an overall increase in the expansion can be observed. The data range bars represent the minimum and maximum achieved values at a given time, and the standard deviation for all measurements ranges from 0.021% to 0.035%. An expansion level of 0.05% was achieved at 27 days, while 43 days were observed to produce an expansion level of 0.12%. Interestingly, a plateau was reached between 217 and 328 days, at 0.23% of expansion, after which an increase in expansion occurred, reaching 0.27% after 546 days.

### 3.2. Shear Resistance Loss

The direct shear resistance is shown in Figure 5, where each point represents the average of three specimens. A slight loss of shear resistance can be observed at 0.05% of expansion from initially 5.35 MPa to 4.18 MPa, after which the shear resistance increased at a constant linear rate up to 4.53 MPa at 0.20% of expansion. However, by visual comparison of the data range bars, the difference between the tests for each expansion level is not evident (extremities overlapping). This was further verified through a one-way ANOVA, which showed that there wasn’t a significant difference between the expansion levels (Table 4).

### 3.3. The Damage Rating Index (DRI)

The damage rating index (DRI) was selected as a tool to quantify the damage features observed at a scale representative of the affected material. Figure 6a,b illustrates the comparison in the DRI numbers and weighted proportions, while Figure 6c,d shows the extended version of the DRI as per [30], obtained in this study. For simplicity, the legend for the bar chart (Figure 6a) only shows the features that were observed in this study (CCA, OCA, and CCP). Traces (less than 1%) of products were observed; however, the distinction between polishing residue and a reaction product couldn’t be made. Two specimens were evaluated per expansion levels; however, the second specimen (labeled “_2”) was evaluated only after the first specimen (labeled “_1”). The decision to test a second specimen was made upon observing inconsistencies in the crack generation and propagation with reference to the expansion level. As such, the second specimens were stored for a longer period of time (i.e., 12 °C), which is considered efficiently capable of mitigating ASR damage [8]. Indeed, this mitigation was not effective for concrete affected by ACR, which can be further observed in [8], where a non-negligeable DRI number was observed for specimens considered sound made with the Kingston-reactive aggregate.

At 0.05% of expansion (DRI numbers of 316 and 779), sharp cracks were present within the aggregate as both closed and opened, which is in accordance with [30]. A large portion of cracks in the cement paste (CCP), however, were observed, which is considered abnormal for a concrete affected by ASR (from 58–78%). However, at 0.05% of expansion, the cracks in the cement paste and the cracks in the aggregate were not necessarily linked to one another, indicating that two mechanisms may have been occurring, which merits further analysis. At 0.12% of expansion (DRI numbers of 478 and 634), an increase was observed, namely for the cracks in the cement paste, whereas open cracks in the aggregates remained similar (from 13–17% of the total contributions towards the DRI calculation). At 0.20% of expansion (DRI numbers of 336 and 708), no significant increase in the open cracks in the aggregate was observed (from 17–25%), with the greater portion of cracks being in the cement paste.

Another approach to visualizing the crack counts by adopting the concept of the extended version of the DRI, as proposed by [8], is illustrated in Figure 7, in which the bars represent the features’ counts. Uncertainty bars of ±10% were further added to these counts to consider operator variability and further differentiate the results. In both cases, for the 0.05% expansion level, the closed cracks in the aggregates (CCA) were similar (207 and 205 counts/100 cm^2^), as observed by the overlapping means. However, these counts decreased for the two other expansion levels. Interestingly, the counts of the cracks in the cement paste (CCP) were very similar for each of the specimens labeled “2”, which was the specimen that was conserved and later tested (i.e., 198, 174, and 183 counts/100 cm^2^ for expansions of 0.05%, 0.12%, and 0.20%, respectively).

### 3.4. Apparent Porosity

Figure 8a,b shows the apparent porosity and absorption, respectively, of the affected concrete with respect to the expansion level achieved. The absorption was calculated using the dry and surface-saturated dry (SSD) masses over the dry mass. Interestingly, both decreased as a function of expansion, whereby the apparent porosity for a sound concrete (assuming that AAR was inhibited when stored at 12 °C for 47 days) was 9% and linearly decreased to 6.3% at 0.20% of expansion. The same trend was indeed observed for the absorption, whereby the absorption decreased as a function of expansion.

Through a one-way ANOVA, it was determined that there was a significant difference between the expansion levels while using the porosity measurements to compare the results (Table 5).

## 4. Discussion

### 4.1. What Does the Multi-Level Assessment Reveal about Damage Due to ACR?

A summary from [8] is shown in Table 6, and Figure 9 illustrates those results to depict ACR-affected concrete in comparison to ASR-affected concrete, as captured through the multi-level assessment. In general, the ACR-affected concrete showed a similar mechanical behavior to ASR-affected concrete for each case (Figure 9a–e); the results from the concrete made with the reactive Kingston aggregate fell within the range of expected values per expansion level. However, the DRI number as a function of expansion (Figure 9f) shows that the Kingston aggregate concrete was above the range, thus highlighting its abnormality when compared to ASR-affected concrete.

Figure 9a,b shows the compressive and tensile strength reductions as a function of expansion, respectively. The compressive strength reduced at a relatively constant rate up to 35% at 0.30% of expansion, while the tensile strength was mostly lost at the beginning of the expansion, where a drop of 45% was observed at 0.05% of expansion. Noticeably, the loss in tensile strength was greatest at 0.20% of expansion, yet the largest difference was between 0% and 0.05% of expansion.

Figure 9c–e shows the SDT outputs as modulus of elasticity reductions, SDI, and PDI as a function of expansion, respectively. The SDI values increased with expansion in a concave shape, where values between 0.13 and 0.25 were obtained from 0% to 0.30% of expansion. The PDI values were somewhat more linear as a function of expansion, with the exception of the PDI at 0.05% of expansion being close to that at 0.12% of expansion (i.e., 0.15 and 0.17, respectively). Moreover, the greatest loss in modulus of elasticity was observed at 0.20% at 43% reduction.

The DRI values for the Kingston aggregate concrete per expansion level (i.e., 0%, 0.05%, 0.12%, and 0.20%) were above the range of values obtained for concrete affected by ASR, even at 0% of expansion (i.e., DRI of 350). When observing the results from [8], one notices that at 0% of expansion, a large portion of cracks in the cement paste had been generated, indicating that there was a mechanism affecting the concrete while being stored at 12 °C (previously verified to inhibit ASR damage [8]. The differences therefore observed in this current study and that of [8] are likely due to the storage conditions being unable to suppress ACR damage, since time intervals between the original storage conditions (i.e., 38 °C and 100% relative humidity) and the storage at 12 °C until testing occurred could have significantly varied. Both studies used the same aggregate (from the same storage) and proportions of mixtures. Nevertheless, it is recommended that a time-based approach be used when assessing concrete affected by ACR due to these differences in results.

In a previous study evaluating the ability of the direct shear test to capture damage due to AAR, a range of values for the shear strength reduction was determined per expansion level [27]. Figure 10 illustrates that the Kingston reactive coarse aggregate continued to present a distinct behavior. As aforementioned, the direct shear test did not highlight significant differences between the expansion levels in this study. The direct shear test, however, measures the concrete’s ability to resist a sliding action between two surfaces. For ASR-affected concrete, the aggregate interlock is diminished since ASR propagates from within the aggregates. Based on the results from the DRI, ACR-affected concrete produces a larger portion of cracks in the cement paste, which does not necessarily affect the aggregate interlock of the concrete.

### 4.2. Common ACR Features and Physical Property

An abundance of micrographs and images depicting the alkali–silica reaction (ASR), the most commonly studied and encountered form of AAR, has been shown in the literature; however, since ACR is a less-understood mechanism, its appearance under a stereomicroscope at 15–16× magnification can likely help to distinguish ACR features from others should they be present in concrete under analysis through the DRI. This section, therefore, aims to present some of the frequently observed features that may indicate damage due to ACR. It is important to note that the analysis used to collect the data for the DRI calculation allows the operator to visualize the spread of the damage at a scale most representative of the affected material when an internal swelling reaction occurs.

At 0.05% of expansion, evidence of ACR was observed through carbonate halos, where a white deposit was seen at the aggregate boundary (Figure 11a). Likewise, the ACR signatures became more prominent at 0.12% of expansion (Figure 11b), with more frequent carbonate halo observations and cracks in the aggregate–paste interface, linking open cracks in the aggregates to each other through the cement paste. Figure 11c shows a crack in the aggregate–paste interface where a halo was also observed, without an association to cracks in the aggregate. At 0.20% of expansion (Figure 11d), cracks in the aggregate were propagating from the aggregate to the cement paste, with a white discoloration at the sides of the cracks, which was observed in Locati et al. [32]. Interestingly, the cracks in the cement paste also showed the white edges along the crack, with some slight white discolouration in the open crack. Cracks in the aggregate–paste interface were more often observed, as well as in the bulk cement paste, with and without association to the cracks in the aggregate (Figure 11e,f). Moreover, cracks appeared wider as the expansion increased. Cracks generated in the aggregate appeared empty of any residue, indicating expansion from such cracks.

The apparent porosity presented an interesting behavior when compared to the porosity taken from an ASR-reactive coarse aggregate (i.e., Springhill–Greywacke), where the porosity tended to increase with expansion for ASR-affected concrete [27], while a decrease was observed for the Kingston aggregate concrete. It is possible that the carbonate halo decreases the apparent porosity of the concrete, which is observed when concrete is subjected to carbonation by calcite filling the pores, due to calcite’s increase in volume of 12% [33], and may further explain why cracks are observed in the interfacial transition zone (ITZ), which is known to be the weakest point in the concrete, resulting in the observed expansions and detachments. Further research is necessary, however, to confirm such a phenomenon.

### 4.3. Sample Size Used for Microscopy Evaluation

Due to reservations about the use of the DRI as an objective method to measure damage, a plot illustrating the cumulative DRI as a function of the sample size (number of analyzed squares) is presented in Figure 12. This plot helps establish the required surface area to be analyzed to obtain a representative sample. Generally, one may identify a sample as the number of physical specimens or cores tested. However, for an analysis in which features are counted, weighted, and their relative proportions compared, the area of analysis used to represent the overall captured damage must be established. The DRI therefore converges towards a mean after a certain number of squares are analyzed. It is to be noted that this convergence refers to the “law of large numbers”, and the sample size (n) at which the DRI converges differs for each test. Nevertheless, this plot serves as a starting point to objectify the DRI as a damage evaluation tool and provides some transparency in the results presentation.

## 5. Conclusions

This study showed that the Kingston aggregate concrete does not behave as purely an ASR-affected concrete when observing its cracking behavior. The following highlights from this study are thus presented:The mechanical responses using the multi-level assessment conducted by [8] show that the damage due to ACR is similar to that of ASR; however, the DRI numbers as a function of expansion were found to be above the range of expected values for ASR. The cracking pattern varies significantly from that of ASR, suggesting ACR as a distinct mechanism. Cracks in the cement paste were found to be the dominant damage feature.The direct shear resistance loss was not captured throughout the expansion levels, nor were the differences between the expansion levels, likely due to the crack propagation being within the cement paste as opposed to within the aggregate.The apparent porosity showed a statistically significant decrease with expansion, which can be attributed to the carbonate halos further reducing the porosity in the aggregate–paste interface. These haloes were frequently observed through the stereomicroscope at 16x magnification, along with cracking of the interface. A study is currently being conducted to better understand the role of the halos with respect to concrete deterioration.

Concrete affected by ACR is less widespread than that affected by ASR; however, with the rise of new efficient and sustainable materials and the unprecedented demand to build concrete infrastructure, it is imperative to provide practitioners with micrographs of concrete affected by ACR while using a readily available, practical, and accessible (low initial and operation costs) evaluation tool. Further research is necessary to understand the cause of this distinct damage through a parametric study using various ACR-susceptible aggregates.

## Figures and Tables

**Figure 1 materials-17-00166-f001:**
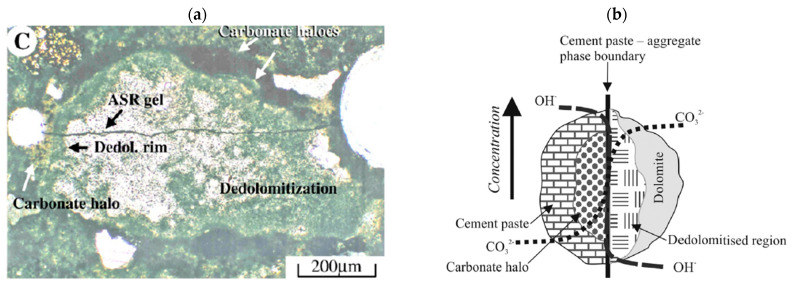
The so-called ACR in concrete, as represented by (**a**) Katayama [1], with the inclusion of ASR, and (**b**) Štukovnik et al. [7], without ASR.

**Figure 2 materials-17-00166-f002:**
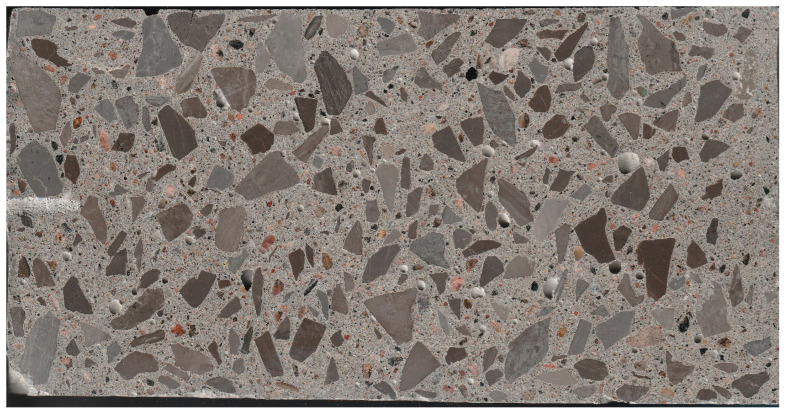
Surface preparation for DRI analysis. The image represents 10 cm by 20 cm.

**Figure 3 materials-17-00166-f003:**
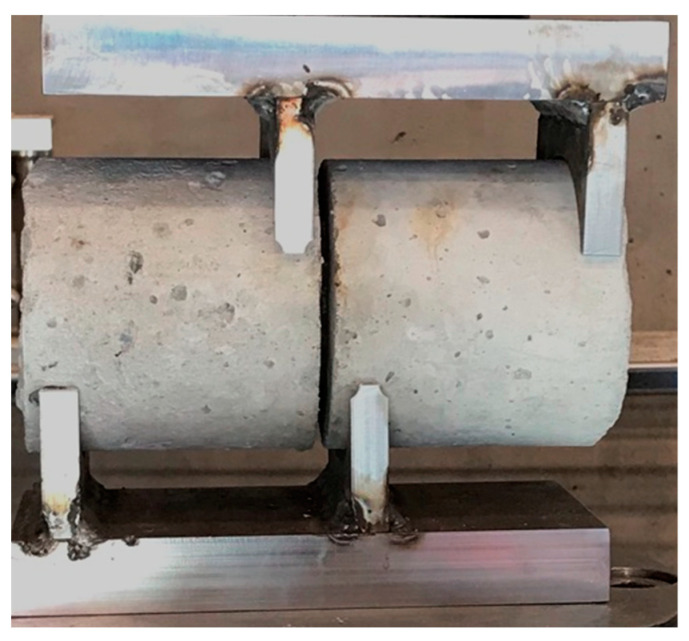
Direct shear set-up, where a notched 10 cm by 20 cm cylindrical specimen is placed in-between the apparatus.

**Figure 4 materials-17-00166-f004:**
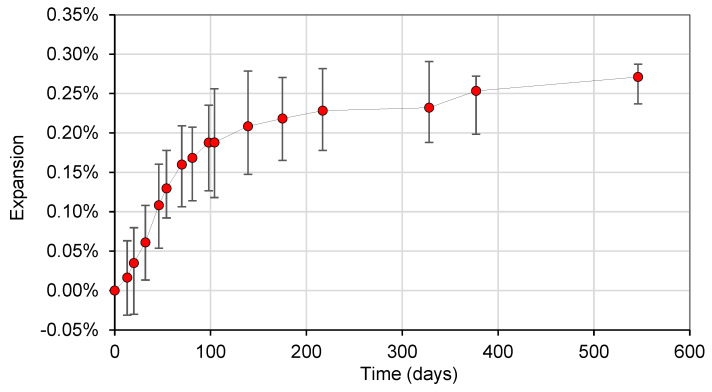
Expansion as a function of time.

**Figure 5 materials-17-00166-f005:**
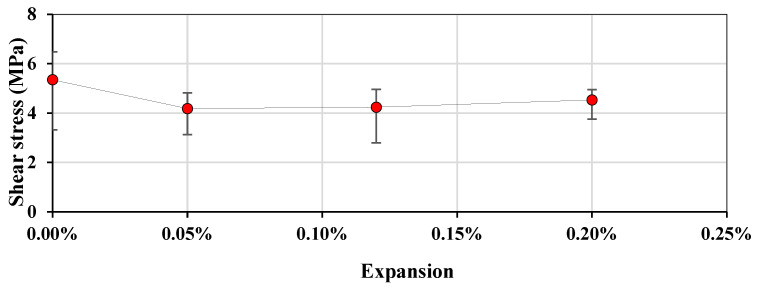
Shear resistance per expansion level.

**Figure 6 materials-17-00166-f006:**
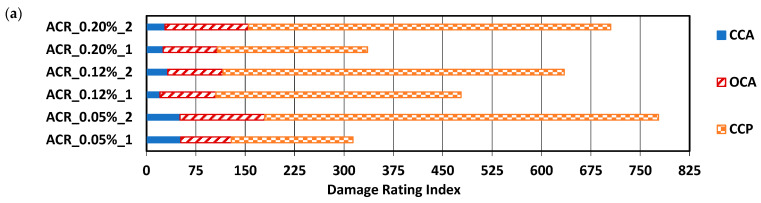
DRI results presented as: (**a**) weighted counts/100 cm^2^, (**b**) weighted proportions, (**c**) unweighted counts/100 cm^2^, and (**d**) unweighted counts as proportions.

**Figure 7 materials-17-00166-f007:**
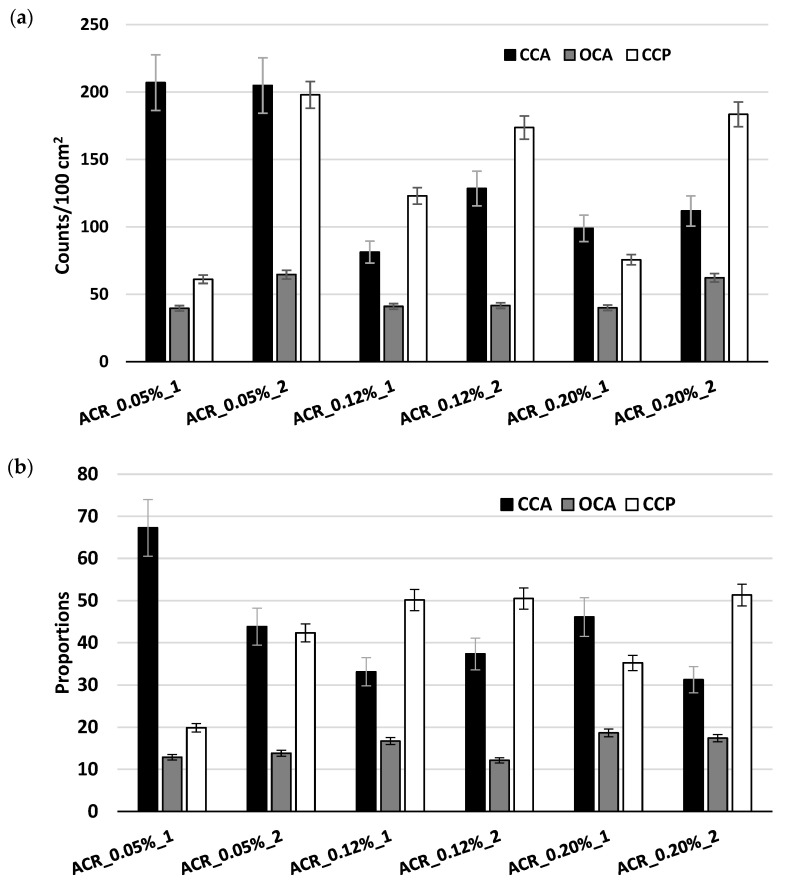
Another approach to the extended version of the DRI as (**a**) counts/100 cm^2^ and (**b**) proportions.

**Figure 8 materials-17-00166-f008:**
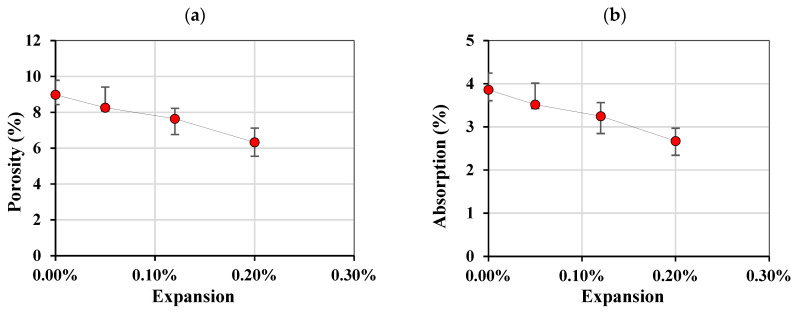
(**a**) Porosity and (**b**) absorption of concrete affected by AAR.

**Figure 9 materials-17-00166-f009:**
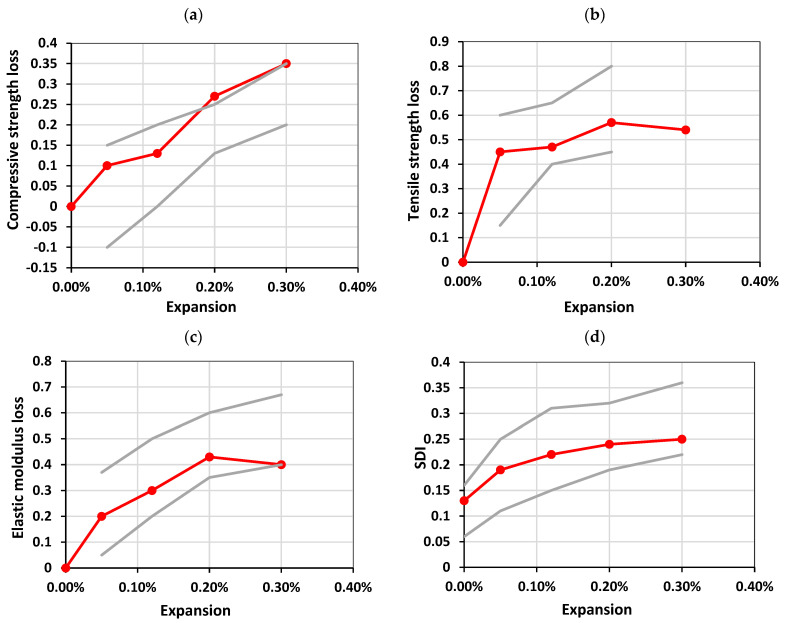
Results from the multi-level assessment per Sanchez et al. [8]: (**a**) compressive strength loss, (**b**) elastic modulus loss, (**c**) tensile strength loss, (**d**) SDI, (**e**) PDI, and (**f**) DRI. The circular tick-marked line represents the values obtained from the Kingston aggregate, while the light solid lines represent the limits from ASR.

**Figure 10 materials-17-00166-f010:**
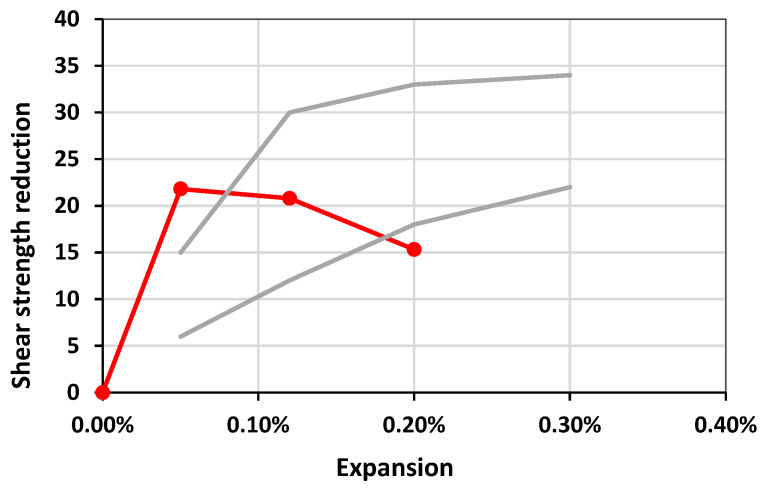
Comparison of shear strength loss from [27] with the results of this current study. The circular tick-marked line represents the values obtained from the Kingston aggregate, while the light solid lines represent the limits from ASR.

**Figure 11 materials-17-00166-f011:**
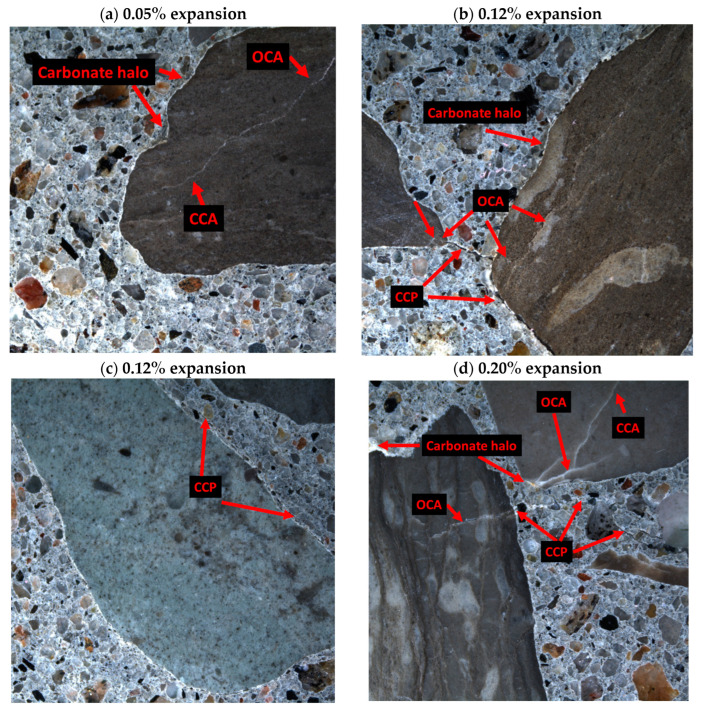
Micrographs of Kingston aggregate concrete at various expansion levels. The field of view represents 1 cm^2^.

**Figure 12 materials-17-00166-f012:**
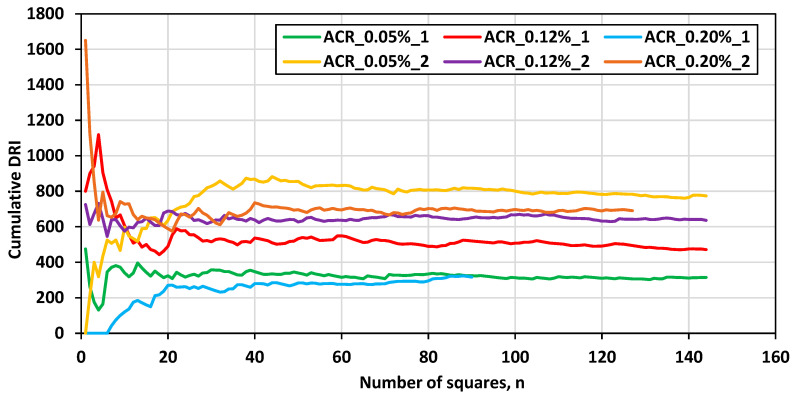
Cumulative DRI vs. sample size.

**Table 1 materials-17-00166-t001:** Aggregate characterization.

Material	Location	Rock Type	Specific Gravity (g/cm^3^)	Absorption (%)	CPT-365 Days, Expansion (%)	AMBT–14 Days, Expansion (%)
Non-reactive fine aggregate	Bracebridge, Ontario (Canada)	Orthoclase, Quartz, Cristoballite, Albite, Bytowmite, Cordierite, Illite, Muscovite, Larnite	2.73	0.37	0.018	0.027
Reactive coarse aggregate	Kingston, Ontario (Canada)	Dolomitic argillaceous limestone	2.61	0.63	0.232 [25]	0.110 [26]

**Table 2 materials-17-00166-t002:** Mix-design.

Cement	Water	Non-Reactive Natural Sand	Reactive Coarse Aggregate		
			4.75–9.5 mm	9.5–12.5 mm	12.5–19 mm
kg/m^3^					
420.00	180.00	766.10	326.78	326.78	336.68

**Table 3 materials-17-00166-t003:** DRI distress features and weighting factors.

Distress Feature	Weighting Factor [26]
Closed cracks in the aggregate, CCA	0.25
Open cracks in the aggregate without or with reaction product, OCA/OCA_RP_	2
Disaggregated/corroded particle, DAP	2
Cracks in the cement paste without or with reaction product, CCP/CCP_RP_	3
De-bonded aggregate, CAD	3

**Table 4 materials-17-00166-t004:** One-way ANOVA for the direct shear test.

Source of Variation	SS	df	MS	F	*p*-Value	F Critic	F > F Critic?	*p*-Value ≤ 0.05?
Between Groups	2.61	3	0.87	0.59	0.64	4.07	no	no
Within Groups	11.86	8	1.48					
Total	14.47	11						

**Table 5 materials-17-00166-t005:** One-way ANOVA for the porosity measurements.

Source of Variation	SS	df	MS	F	*p*-Value	F Critic	F > F Critic?	*p*-Value ≤ 0.05?
Between Groups	34.20	3	11.40	49.77	3.74 × 10^−12^	2.90	yes	yes
Within Groups	7.33	32	0.23					
Total	41.53	35						

**Table 6 materials-17-00166-t006:** Summary of results in comparison with ASR.

Reference Expansion Level (%)	Compressive Strength Loss	Tensile Strength Loss	SDI	Stiffness Loss	Shear Strength Loss	Apparent Porosity (%) [31]	DRI
	ASR						
0.00–0.03	-	-	0.06–0.16		-	-	100–155
0.04 ± 0.01	−10 to 15%	15 to 60%	0.11–0.25	5 to 37%	6 to 15%	5.2	210–440
0.11 ± 0.01	0 to 20%	40 to 65%	0.15–0.31	20 to 50%	12 to 30%	-	330–500
0.20 ± 0.01	13 to 25%	45 to 80%	0.19–0.32	35 to 60%	18 to 33%	6.72	500–765
0.30 ± 0.01	20 to 35%		0.22–0.36	40 to 67%	22 to 34%	-	600–925
	Kingston aggregate (ACR)						
0.00–0.03	-	-	0.13	-	-	8.98	350
0.04 ± 0.01	10%	45%	0.19	20%	22%	8.25	575
0.11 ± 0.01	13%	47%	0.22	30%	21%	7.64	885
0.20 ± 0.01	27%	57%	0.24	43%	15%	6.33	900
0.30 ± 0.01	35%	54%	0.25	40%			910

## Data Availability

Data are contained within the article.

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
