# Peer review of "Damage Generated and Propagated by the AAR Reactive Aggregate from Kingston, Ontario, Canada"

_materials, 2023, doi:10.3390/ma17010166_

Round 1

Reviewer 1 Report

Comments and Suggestions for Authors

Please go through the comments below to improve the manuscript:

·      Please provide a more quantitative description of the results in the Abstract. Currently, the Abstract is a little too short and most of the sentences discuss the background and methods.

·      The title can be shortened by eliminating “A contribution to understanding”.

·      Line 29: remove “of which”.

·      Lines 44-47: The sentence is too long, please consider separating it into two sentences.

·      Please add the scale bar to Figures 2, 3, and 11.

·      Section 2 – Scope of the Work can be integrated into the Introduction.

·      The wording “ACR susceptible reactive Kingston coarse aggregate” seems a bit long and confusing.

·      Line 105: This sentence reads like incomplete.

·      Figure 4: Please move the horizontal axis down to cross the vertical axis at -0.05%

·      Figure 5: How many specimens were tested for each point? 

·      Section 4.4: How does the “absorption” be tested and measured?

·      Figures 9 and 10: Please add the legend for all the plots. 

Author Response

The authors would like to thank the reviewer for their time and contribution to the publishing process of this research work. All comments on improvements were addressed and highlighted in the manuscript. Please find the comments from the corrections to the manuscript below:

Review comment

Addressed by contributing author

Please provide a more quantitative description of the results in the Abstract. Currently, the Abstract is a little too short and most of the sentences discuss the background and methods.

More emphasis was added onto the main findings of this research. Thank you kindly for this constructive comment.

·      The title can be shortened by eliminating “A contribution to understanding”.

·      Line 29: remove “of which”.

·      Lines 44-47: The sentence is too long, please consider separating it into two sentences.

·      Please add the scale bar to Figures 2, 3, and 11.

·      Section 2 – Scope of the Work can be integrated into the Introduction.

·      The wording “ACR susceptible reactive Kingston coarse aggregate” seems a bit long and confusing.

·      Line 105: This sentence reads like incomplete.

·      Figure 4: Please move the horizontal axis down to cross the vertical axis at -0.05%

All of these issues were corrected.

·      Please add the scale bar to Figures 2, 3, and 11.

·      Figures 9 and 10: Please add the legend for all the plots. 

The specimen sizes were added to figure captions 2 and 3. Figure 11 already has the field of view of 1 cm2 in the caption.

The legend was added as a caption since only one type of concrete was compared to the ASR range.

·      Figure 5: How many specimens were tested for each point? 

·      Section 4.4: How does the “absorption” be tested and measured?

3 specimens tested per point, a line was added to the text to specify.

Absorption is measured by taking the difference of the dry and SSD masses over the dry mass. This was added to the text for clarity.

Reviewer 2 Report

Comments and Suggestions for Authors

How prevalent is ACT? The study presents an interesting case and might even hint that ACT is under reported. I think the manuscript would greatly benefit from providing some perspective on the issue instead of simply limiting the study to Ontario Canada.

Overall, good work

Author Response

Thank you kindly for your time and effort during this review. You are correct, we have many cases of ASR however, we may have been misdiagnosing some of the damages that could have been ACR or other mechanisms like DEF that are less reported and understood. The goal of this paper was to shed some light that there was differences ASR vs ACR. I will add a line regarding your second comment, since we did not want to generalize ACR as we were only looking into the Kingston aggregate found in Ontario, Canada. This aggregate was referenced in other literature and other ACR susceptible aggregates may present differences since the mechanism in not yet well understood. 

Reviewer 3 Report

Comments and Suggestions for Authors

Comments on the Quality of English Language

Corrections are needed.

Author Response

The authors would like to thank the reviewer for their time and contribution to the publishing process of this research work. All comments of improvements were addressed and highlighted in the manuscript. Please find the comments from the corrections to the manuscript below:

Review comment

Addressed by contributing author

1. It seems that the paper only focused on Ontario, Canada. Can it be extended to further regions with different climates?

Thank you for this comment. As it is now, since the mechanism of ACR remains unknown, the authors did not want to generalize to other ACR susceptible aggregates. A study evaluating a wide range of ACR susceptible aggregates would be required to understand the effect in various climates. A line was added to the text for clarity on this point.

2. Introduction is written ambiguously and has very long paragraphs. It is recommended to have a more concise Introduction by focusing on the paper's achievements.

The ambiguity was reviewed and updated to be more concise and redirect the focus to the evaluation tools to capture damage.

3. A simple section (Section 2) is allocated to the paper scope. Please merge it with Section 1.

This was combined.

4. As I understood, you only study the concrete elasticity. Does the concrete have a plastic behavior?

The plasticity is also evaluated through the plastic deformation index (PDI) from Figure 9e. However, the specimens are loaded to 40% of the f’c of an undamaged concrete which tests the material at the elastic range without causing more damage.

5. Can you please provide details about aggregate dimension (e.g., radius)?

The details were provided in Table 2 which is a standard mixture for AAR testing.

6. Conclusions are too long and not informative. Please focus on the paper's achievements

The conclusions were adjusted, thank you for this constructive comment.

7. The paper is heavily self-cited (around one-third) which is unacceptable. The authors are

suggested to discuss the recently published papers on damage models in porous media and concrete. The following paper should be addressed by authors to discuss the optimization methods.

This was addressed while reworking the introduction.

• Probabilistic failure mechanisms via Monte Carlo simulations of complex microstructures

https://doi.org/10.1016/j.cma.2022.115358

Thank you for this suggestion.

Round 2

Reviewer 3 Report

Comments and Suggestions for Authors

Accept.

Comments on the Quality of English Language

fine